# Carbon Quantum Dots from Pomelo Peel as Fluorescence Probes for “Turn-Off–On” High-Sensitivity Detection of Fe^3+^ and L-Cysteine

**DOI:** 10.3390/molecules27134099

**Published:** 2022-06-25

**Authors:** Dianwei Zhang, Furui Zhang, Yonghong Liao, Fenghuan Wang, Huilin Liu

**Affiliations:** 1School of Light Industry, Beijing Technology and Business University, 11 Fucheng Road, Beijing 100048, China; zhangdianwei@btbu.edu.cn (D.Z.); zhfree1918@163.com (F.Z.); 2School of Food and Health, Beijing Technology and Business University, 11 Fucheng Road, Beijing 100048, China; liuhuilin@btbu.edu.cn

**Keywords:** carbon quantum dots, turn-off–on, pomelo peel, fluorescence probe, Fe^3+^, L-cysteine

## Abstract

This study designed a “turn-off–on” fluorescence analysis method based on carbon quantum dots (CQDs) to detect metal ions and amino acids in real sample systems. CQDs were derived from green pomelo peel via a one-step hydrothermal process. The co-doped CQDs with N and S atoms imparted excellent optical properties (quantum yield = 17.31%). The prepared CQDs could be used as fluorescent “turn-off” probes to detect Fe^3+^ with a limit of detection of 0.086 µM, a linear detection range of 0.1–160 µM, and recovery of 83.47–106.53% in water samples. The quenched CQD fluorescence could be turned on after adding L-cysteine (L-Cys), which allowed detection of L-Cys with a detection limit of 0.34 µM and linear range of 0.4–85 µM. Recovery of L-Cys in amino acid beverage was 87.08–122.74%. Visual paper-based testing strips and cellulose/CQDs composite hydrogels could be also used to detect Fe^3+^ and L-Cys.

## 1. Introduction

Since their initial discovery in 2004, carbon quantum dots (CQDs) have drawn great attention and been widely used as novel carbon-based nanomaterials. Typically, most CQDs are spherical nanoparticles; the average diameter is usually less than 10 nm [1]. In the past 10 years, CQDs have attracted great attention because of their superior properties of low toxicity, good biocompatibility and light resistance, easy surface modification, and environmental friendliness [2], providing good potential for use in applications, such as drug delivery, sensing, biological imaging, fluorescent inks, catalysis, and light-emitting diodes [3,4,5]. The obvious advantages of green chemistry have stimulated the great potential of CQDs in synthesis and application, making them an indispensable part of carbon nanomaterials. To date, many CQDs have been synthesized from biomass, such as candle root, Prunus avium fruit, and garlic [6,7,8]. Recently, heteroatom-doped CQDs have gradually emerged into the field of carbon-based nanomaterials because they can effectively adjust the composition of surface groups and change optical properties [9,10]. However, the quantum yield (QY) of one or two heteroatom-doped CQDs from biomass materials is very low, which complicates the practical application of CQDs. Therefore, there remains an urgent need for methods of large-scale preparation of high-QY biomass CQDs.

Currently, CQDs with optosensing capabilities are attracting widespread attention from researchers. In consideration of the fluorescence intensity changes in CQDs during the analysis and detection process, CQDs have been used as sensitive analytical fluorescent nanoprobes to detect certain metal ions (Cu^2+^, Ag^+^, Fe^3+^, Hg^2+^) and amino acids [11,12,13,14]. Such applications are important because low levels of some heavy metal ions may cause serious physiological problems in the human body [15]. In particular, Fe^3+^ is abundant in the human body as an essential element. L-Cysteine (L-Cys) is a common and significant amino acid in living organisms and the only one among more than 20 common amino acids that contains a thiol moiety. Because of its important physiological functions, it has promising application prospects in biopharmaceuticals, the food industry, animal feed, and the cosmetics industry [16,17]. The traditional detection methods of Fe^3+^ and L-Cys were mainly large-scale instrument chromatographic analysis methods [18,19,20]. However, in general, these methods have several limitations of time-consuming pretreatments, complex operations, complicated detection processes, and relatively expensive instruments. Thus, there remains a need for a facile, rapid, inexpensive, highly sensitive, and selective analytical method for simultaneous analysis of Fe^3+^ and L-Cys. Recently, Huang et al. prepared red-emitting CQDs co-doped with nitrogen–boron–sulfur to detect Ag^+^ and L-Cys in living cells and complex biological fluids [11]. Yan et al. constructed a simple sensing system based on fluorescence switching of CQDs for Hg^2+^ and L-Cys analysis [12]. Zong and co-workers developed an “off–on” fluorescent CQD probe to detect Cu^2+^ and L-Cys in aqueous solution [13]. Recently, Sun’s group achieved a sensitive “on–off–on” model for Hg^2+^ and L-Cys detection using nitrogen–sulfur (N-S) co-doped CQDs prepared from gardenia fruit [21]. However, the low toxicity and high QY of CQDs is still a problem that we urgently need to improve.

Herein, we describe the innovative design of CQDs with a specific and sensitive fluorescence “turn-off–on” mode, which could be used for sensitive and simple determination of Fe^3+^ and L-Cys. N-S co-doped CQDs were prepared via a one-step hydrothermal method using waste green pomelo peel as a precursor. The prepared N–S co-doped CQDs could be used as efficient fluorescent “turn-off” probes for Fe^3+^ detection with high selectivity, low LOD, and wide linear detection range. Interestingly, the quenched fluorescence could be “turned-on” after adding L-Cys, which could provide a high-selectivity and -sensitivity detection method for L-Cys in real samples. In addition, visual paper-based testing stripes and cellulose/CQDs composite hydrogels with N–S co-doped CQDs had also been used for Fe^3+^ and L-Cys detection in the solid state according to the change in color and off–on fluorescence.

## 2. Results and Discussion

### 2.1. Optimization of the Preparation Conditions of CQDs

The detection pathway for Fe^3+^ and L-Cys based on CQDs’ fluorescent switch is shown in Figure 1. Carbon sources play a very significant role in the preparation of the CQDs and there are many carbon sources for the preparation of CQDs. So far, CQDs synthesized from different carbon sources have been reported. Taking into account the low toxicity of CQDs and the cost of source, biomass and its derivatives have attracted extensive attention because of their outstanding performance in availability, reproducibility, and low cost. Navel orange peel, orange peel, and pomelo peel from the citrus genus of Rutaceae were used as the carbon sources for the CQDs preparation in the study (Appendix A). It was interesting that CQDs derived from pomelo peel had better fluorescence intensity than navel orange peel and orange peel (Appendix A). In addition, the effects of different solvents on CQDs’ fluorescence intensities were conducted to explore by H_2_O, C_2_H_6_O, CH_3_COOH, and phosphate buffer solution (PBS). It is obvious from Appendix A that the CQDs have the highest fluorescence response in aqueous solution. Then, the optimum concentration of CQDs in aqueous solution was optimized (Appendix A). The CQDs aqueous solution (0.01 mg mL^−1^) obtained under the optimized conditions was used for further study.

### 2.2. Characterization

Figure 1A showed the particle sizes and shapes of CQDs, which were obtained by TEM analysis. The TEM images showed that the prepared CQDs were uniformly distributed and could be well dispersed in aqueous solutions without apparent aggregation. High-resolution TEM images further exhibited the CQDs with a graphitic structure of clear lattice fringes about 0.21 nm. Figure 1B shows that the synthesized CQDs were not very uniform in shape, with a size range of about 1.2–8.4 nm. The average particle size of CQDs was about 5.5 nm based on 100 random nanoparticles. Typical XRD patterns of CQDs (Figure 1D) displayed highly disordered carbon structure with a center of broad diffraction peak at 22.1° [22].

The functional group information of CQDs was obtained through FTIR spectroscopy (Figure 1C). The strong bands at 3442 cm^−1^ and 2924 cm^−1^ correspond to the N–H and –CH_2_ group characteristic vibrational bands on the surface of CQDs, respectively [23,24]. The peaks at 1520–1720 cm^−1^ were attributed to the C=O groups’ stretching vibration [25]. An obvious band at 1403 cm^−1^ was derived from the COO^−^ stretching vibration [26]. The band at 1260 cm^−1^ was attributed to C–N groups [27]. A stretching vibration peak at 1078 cm^−1^ resulted from –SO_4_^2−^ bands [28]. All these absorption peaks indicated that N and S were successful at doping on the CQD surface.

The elemental composition of CQDs was determined by XPS (Figure 1E). There were four obvious peaks of 532.5, 399.9, 284.9, and 186.5 eV on the XPS spectrum, representing the presence of O_1s_, N_1s_, C_1s_, and S_2p_ on the CQD surface. According to the XPS data, the elemental composition of the CQDs was 67.96% C, 4.03% N, 27.81% O, and 0.2% S. As shown in Figure 1F, four peaks at 288, 286.5, 286.1, and 284.7 eV in the high-resolution C_1s_ spectrum, represented O=C–O, C–O/C–N, C–S, and C–C/C=C bonds, respectively [29,30]. As shown in Figure 1G, three peaks appeared at 400.5, 399.8, and 399 eV on the N_1s_ spectrum, implying the pyrrolic N, C–NH_2_, and NH_3_, respectively [31]. There were two main components at 165.4 and 168 eV on the S_2p_ spectrum (Figure 1H), exhibiting that S was on the CQD surface. The deconvolution peaks at 165.4 eV were attributed to –C–S–O bond; other components at 168 eV are attributed to sulfur oxides [32]. The XPS and FTIR spectra features confirmed that N and S were successfully co-doped in the CQDs, and some relevant functional groups existed on the CQD surface, such as amino, carboxyl, carbonyl, and hydroxyl groups.

### 2.3. Optical Properties of CQDs

The UV absorption and fluorescence spectrometry of the CQDs were investigated. As shown in Supporting Information Appendix A, the UV spectrum contained two typical peaks at 282 nm and 317 nm. The former was attributed to the n-π* electronic transition of the C=O bands of carboxyl groups on the surface of CQDs, while the latter was assigned to modification of other components on the CQD surface [33,34]. The fluorescence excitation and emission spectra of CQDs’ aqueous solution are shown in blue and red lines in Appendix A. The optimal emission wavelength of the CQDs was about 410 nm under the excitation wavelength, about 310–350 nm. The CQDs’ aqueous solution was brown under natural light while it exhibited a bright blue fluorescence under the excitation of a UV light at 365 nm (Inserted Appendix A). Accordingly, the excitation wavelength of CQDs was further optimized in Appendix A. The emission wavelength showed no evident shift at 410 nm when the excitation wavelength of CQDs was set in a range of 300 to 360 nm. The fluorescence intensity gradually enhanced and then decreased rapidly, and the inflection point with the greatest fluorescence intensity occurred at 330 nm. Therefore, we chose 330 nm as the maximum excitation wavelength (Appendix A).

The optical stability of the prepared CQDs was further evaluated at ambient temperature and under UV light to evaluate the potential practical application. The preliminary experimental results showed that the CQDs had good fluorescence stability after storage at ambient temperature for 30 days (Appendix A). In addition, the prepared CQDs had good resistance to photobleaching, the fluorescence intensity of CQDs was almost unaffected after 90 min of continuous irradiation at 365 nm (Appendix A). The CQDs also maintained constant fluorescence intensity over a broad pH range, increasing from 3 to 10 (Appendix A). The salt tolerance of CQDs was also investigated, in ultrahigh-concentration NaCl (up to 1.0 M) solution, and the CQDs still had strong fluorescence intensity, indicating that CQDs possess the advantage of salt tolerance (Appendix A). The absolute QY of the CQDs reached 17.31% (Appendix A). Therefore, the prepared CQDs still had good stability under extremely harsh conditions, which was beneficial to the potential applications in sensing and analysis.

### 2.4. Selectivity and Sensitivity

There were fifteen common metal ions used in the interference experiments to evaluate the selectivity and sensitivity of CQDs. The fluorescence intensities of CQDs containing different metal ions are studied in Figure 2A. In order to evaluate the specificity and feasibility of prepared CQDs for the detection of Fe^3+^, the fluorescence responses of various potentially interfering metal ions to the CQDs/Fe^3+^ system at a constant concentration were investigated (Figure 2B). The concentration of other metal ions was twice and five-times that of Fe^3+^, as shown in Appendix A. These studies showed that only Fe^3+^ could effectively decrease the fluorescence intensity of the CQD aqueous solution. This more prominent selectivity for Fe^3+^ is possibly caused by the much stronger interaction of Fe^3+^ with the CQDs than the other 14 cations. The result was intriguing, which gave us some inspiration for Fe^3+^ detection using the “turn-off” CQDs. However, after introducing L-Cys into the CQDs/Fe^3+^ system, the fluorescence intensity was recovered. Other amino acids were also tested at the same time, and it was apparent that their fluorescence recovery extent was inferior to L-Cys (Figure 2C). The selectivity of homocysteine and glutathione based on the CQDs/Fe^3+^ system for L-cysteine was also studied (Appendix A). Accordingly, we designed a “circular pattern” by using CQD solutions containing 15 common metal ions (Figure 2D). Under UV light, the white circular pattern in daylight exhibited a blue fluorescence except in the presence of Fe^3+^. Thus, the CQDs/Fe^3+^ system could be successfully applied to L-Cys analysis as a “turn-on” fluorescence probe. The changes in CQD fluorescence intensity during the proposed sensing process is shown in Figure 1.

### 2.5. Optosensing of Fe^3+^ and L-Cys

To evaluate the detection performance of the prepared CQDs, fluorescence intensity was used as the evaluation index to reflect various Fe^3+^ concentrations from 0.1 to 160 µM. Figure 3A shows that the fluorescence response of CQDs gradually decreased when the Fe^3+^ concentration increased. In addition, the insert in Figure 3A shows that after irradiation with UV light, the blue light of CQD aqueous solution vanished with the increasing concentration of Fe^3+^. The linear range of Fe^3+^ detection was further investigated by the changes in fluorescence intensity of CQDs (Figure 3B). In a range of 0.1–60 μM, an obvious linear relationship was obtained with a good correlation coefficient (R^2^ = 0.994). The linear equation was fitted as y = 0.00267x + 1.001. The LOD was calculated as 0.086 µM, according to Equation (1) [12,35]:LOD = 3σ/s(1)
where σ is the relative standard deviation of the blank NCD sample recorded (*N* = 10) and s is the slope of the linear calibration curve.

Notably, the fluorescence signal of CQDs was gradually recovered when L-Cys was added from a concentration of 0.4 to 1000 mM (Figure 3C). The insert in Figure 3C shows that after irradiation with UV light, the blue light in the CQDs/Fe^3+^ aqueous solutions obviously recovered by increasing the content of L-Cys. Figure 3D shows the correlation of the relative fluorescence intensity of CQDs with a concentration of L-Cys from 0.4 to 1000 mM. The relative fluorescence intensity of N/S-CQDs showed a good linear relationship with the concentration of L-Cys between 0.4 and 85 µM (Inserted in Figure 3D). An appropriate correlation coefficient was obtained by R^2^ of 0.990. The fitted linear equation is y = 0.00148x + 1.0734. The LOD for L-Cys was 0.34 µM calculated by the same method as Fe^3+^ detection.

In order to further evaluate the practicability of the “turn-off–on” assay, the CQDs were applied to the detection of Fe^3+^ in laboratory tap water and L-Cys in amino acid beverage samples. Neither of the two samples contained the target to be measured. The recovery was calculated as difference before and after adding standard targets of 10, 30, and 60 μM, which was 83.47–106.53% and 87.08–122.74% in Appendix A. The results demonstrated the feasibility of the developed CQD probe for Fe^3+^ and L-Cys detection in a complex matrix. The feasibility, reliability, and application potential of the CQD-based optosensing for detection of targets were demonstrated.

The repeatability of the CQDs/Fe^3+^ system for the L-Cys detection is shown in Appendix A. Six repeated measurements showed no significant differences in the study. That is to say, the established detection method has good reproducibility.

### 2.6. Possible Mechanism of “Turn-Off–On” Mode

The photoluminescence quenching process is complex, usually including static and dynamic quenching. In this study, CQDs displayed a “turn-off–on” mode when adding Fe^3+^ and L-Cys into CQDs aqueous solution. When Fe^3+^ was added, the fluorescence intensity of CQDs was reduced (Figure 1), which may be due to the coordination of S-containing groups of CQDs with Fe^3+^ to form Fe^3+^-S bonds [36]. However, the CQD fluorescence recovered after adding L-Cys into the CQDs/Fe^3+^ mixture, which may be caused by stronger binding interaction between Fe^3+^ and L-Cys, meaning that Fe^3+^ can be desorbed from the CQD surface. The fluorescence-sensing process can be studied with the experiments of UV absorption spectroscopy and fluorescence lifetime measurement. As shown in Figure 4A, the characteristic absorption peak of CQDs was obviously red shifted from 318 to 328 nm in the presence of Fe^3+^, indicating that Fe^3+^ formed a complex with the CQDs. However, under the presence of L-Cys, the absorption peaks for the CQDs recovered to their original positions, which confirmed the removal of Fe^3+^ on the CQD surface. Interestingly, an average fluorescence lifetime for the aqueous CQDs solution was 1.19 ns (Figure 4B), which was essentially the same as the lifetimes for the CQDs/Fe^3+^ system (1.22 ns) and after adding L-Cys to the system (1.20 ns). These results suggested that the fluorescence-quenching mechanism was static fluorescence quenching in these processes. Combined, these results suggested that the developed CQDs could be applied as bifunctional probes for Fe^3+^ and L-Cys recognition.

### 2.7. Visualization Cellulose/CQDs Composite Hydrogels and Test Paper

Cellulose/CQDs composite hydrogels and CQD-based strip-type test papers (Figure 5) were prepared using simple blending and coating processes. The composite hydrogels and CQD-based test papers had good biocompatibility and biodegradability because they were composed of cellulose and CQDs. After the addition of Fe^3+^ (Figure 5), the cellulose/CQDs hydrogels and CQD-based test papers showed color changes and drop-offs in fluorescence. Surprisingly, the cellulose/CQDs hydrogels and CQD-based test papers then showed obvious color recovery with the addition of L-Cys. Therefore, cellulose/CQDs hydrogels and CQD-based test papers could act as optical sensors for Fe^3+^ and L-Cys. Compared with aqueous solutions of CQDs, the CQD-based hydrogels and test papers were easier to operate and use, and, therefore, had broader application prospects. More importantly, both cellulose/CQDs hydrogels and CQD-based test papers were environmentally friendly and disposable. All these findings suggested that the prepared CQDs had great potential for applications as bifunctional sensors for recognition of Fe^3+^ and L-Cys.

## 3. Materials and Methods

### 3.1. Materials

Green pomelo, navel orange, and tangerine were purchased from a fruit shop in Beijing, China. CoCl_2_, Ba(NO_3_)_2_, AgCl, MgCl_2_, CuSO_4_, NaCl, CaCl_2_, ZnCl_2_, CdCl_2_, NiCl_2_, and FeSO_4_·6H_2_O were acquired from Tianjin Chemical Reagent (Tianjin, China). MnCl_2_, FeCl_3_·6H_2_O, and AlCl_3_·6H_2_O were obtained from Aladdin (Shanghai, China). The 15 amino acids used in the study were obtained from Alfa Aesar (Tianjin, China) Chemical. 1-Allyl-3-methylimidazolium chloride (AmimCl) ionic liquid was acquired from Sigma-Aldrich (St. Louis, MO, USA). Dialysis bags (1000 Da molecular weight cutoff, MWCO) were supplied by MYM Biological Technology (Hyderabad, India). All reagents and chemicals used in the experiments were of analytical grade and not further purified. Tap water sample was obtained from our laboratory.

### 3.2. Instruments

All fluorescence spectra were obtained by Synergy H1 full-function microplate reader (BioTek, Winooski, VT, USA). Transmission electron microscopy (TEM, FEI, Thermo Fisher Scientific, Waltham, MA, USA) images were obtained using a Talos F200X electron microscope. Fourier transform infrared (FTIR, Nicolet Instruments, Thermo Fisher Scientific, Waltham, MA, USA) spectra (400–4000 cm^−1^) were obtained on IS10 spectrometer. X-ray photoelectron spectroscopy (XPS, Thermo Fisher Scientific, Waltham, MA, USA) was carried out by ESCALAB 250Xi spectrometer. Fluorescence lifetime was recorded via FLS1000 fluorescence spectrometer (Edinburgh Instruments, Livingston, UK). X-ray diffraction (XRD, Bruker, Karlsruhe, Germany) analysis using a D8 ADVANCE instrument using Cu K_α_ (λ = 0.15405 nm).

### 3.3. Synthesis of CQDs

The CQDs were derived via one-step hydrothermal method using waste green pomelo peel as the carbon source. Dried green pomelo peel was ground into powder in the mortar and 0.5 g of powder was dispersed in deionized water of 50 mL. We transferred the mixture to the polytetrafluoroethylene-lined stainless steel autoclave of 100 mL. After heating for 5 h at 180 °C, we cooled to ambient temperature naturally and the suspension was centrifuged at 10,000 rpm for 10 min to obtain a yellow-brown supernatant. It was then filtered through a 0.22 um cellulose filter membrane and dialyzed for 24 h using deionized water (MWCO: 1000 Da); deionized water was changed every 8 h. Subsequently, the CQDs were obtained by evaporation of the solvent and placed in a vacuum drying oven for further drying.

For comparison, navel orange peel and tangerine peel were selected as alternative raw materials to synthesize CQDs. The same experimental steps were repeated, and the corresponding tests of fluorescence intensity were performed.

### 3.4. QY of CQDs

The QY of the obtained CQDs was determined by comparison method with a reference substance of quinine sulfate (QY = 56% in 0.1 M H_2_SO_4_) [37]. The QY_CQDs_ was calculated according to Equation (2):QY_CQDs_ = QY_R_ × (F_CQDS_/F_R_)(A_R_/A_CQDs_)(η_R_/η_CQDs_),(2)
where F represents the integral of the fluorescence intensity, A is the absorbance at the excitation wavelength, η represents the refractive index of the solvent (1.33 for water), and subscript R represents the reference substance quinine sulfate.

### 3.5. Fluorescence Stability of CQDs

To evaluate fluorescence stability of the CQDs in different solvents, the CQDs were dispersed in deionized water, ethanol, acetic acid, and PBS buffer, respectively. We added aliquots of these solutions (200 μL) to 96-well plates to measure the fluorescence intensity. For a continuous investigation of fluorescence stability, the fluorescence intensity of CQDs was measured every 3 days for 30 days at ambient temperature. The effect of different irradiation time (5, 10, 20, 30, 40, 50, 60, 70, 80, 90 min) with ultraviolet (UV) light was performed in a UV light box at 356 nm. The effect of different pH values on fluorescence intensity was investigated from 3 to 10. Aliquots (100 μL) NaCl solutions of different concentrations (10, 20, 50, 100, 150, 200, 300, 400, 500 mM) were added into 100 μL of CQD solution, respectively, and the fluorescence intensities were recorded.

### 3.6. Selectivity and Interference Measurements of CQDs

In the selectivity experiment, 100 μL of CQD (0.01 mg·mL^−1^) solution was added to the 96-well plate. Aliquots of individual metal ion solution (Mg^2+^, Ag^+^, Zn^2+^, Cd^2+^, Co^2+^, Na^+^, Al^3+^, Fe^2+^, Fe^3+^, Cu^2+^, Ni^2+^, Ca^2+^, Mn^2+^, and Ba^2+^, 100 μL, 1 mM) were, respectively, mixed with CQD solution, then the fluorescence intensity was measured. We then added 100 μL of Fe^3+^ solution to the CQDs solution and measured the fluorescence intensity again. The interference of metal ions on Fe^3+^ detection by CQDs was determined by the relative fluorescence intensities in the CQD solution containing different metal ions before and after adding Fe^3+^ solution.

### 3.7. Sensitive Detection of Fe^3+^ and L-Cys

In the Fe^3+^ detection experiment, 100 μL of Fe^3+^ solutions of different concentrations (0.1–160 µM) was added into aqueous CQD (100 μL, 0.01 mg·mL^−1^) solutions, then 100 μL aliquots of different metal ions (1 mM) were added into the CQD solutions for comparison. Furthermore, the states of the CQDs in Fe^3+^ solutions with different concentrations were recorded under sunlight and UV light.

We added 100 μL of L-Cys solutions (0.4–1 mM) to the mixed solutions of CQDs (100 μL, 0.01 mg·mL^−1^) and Fe^3+^ (100 μL, 4 × 10^−4^ mol·L^−1^) for L-Cys detection. After sufficient mixing, we recorded the fluorescence intensities immediately (equilibration time 10 s). The fluorescence intensities of all samples in the experiment were recorded in a wavelength range of 360–700 nm and the excitation wavelength was 330 nm. The optical behavior of the CQDs in the presence of the mixture was recorded under sunlight and UV light.

Tests were also performed with other amino acids in the mixture of CQDs and Fe^3+^ under the same conditions. Measurements of sensitivity and selectivity were repeated three times.

### 3.8. Fe^3+^ and L-Cys Detection in Real Samples

Fe^3+^ was dissolved into the tap water to give different concentrations of 50, 100, and 400 µM for detection (excitation at 360 nm). For L-Cys detection, the amino acid beverage sample was diluted 100 times and then mixed with CQD solution containing Fe^3+^. Aliquots (200 μL) of 10, 30, and 60 μM mixtures of amino acid beverage sample were added into 96-well plates for detection (excitation at 360 nm).

### 3.9. Preparation of Cellulose/CQDs Composite Hydrogels

The Cotton pulp (2.5 g) was mixed into 100 g of AmimCl and stirred vigorously at 80 °C to obtain a homogeneous solution, then placed in a vacuum oven for 4 h to remove air bubbles. The solution was then poured slowly into a 12-well plate, which was subsequently put into an AmimCl/H_2_O (*v*/*v* = 1/1) coagulation bath for 24 h. We completely removed AmimCl ionic liquid by ethanol/H_2_O (*v*/*v* = 1/1). The prepared cellulose hydrogels were soaked in CQD aqueous solution for 24 h to obtain CQD composite hydrogels.

### 3.10. Preparation of CQD-based Test Papers

The aqueous solution of CQDs was brushed onto strip-type filter papers, which were previously treated with dimethylformamide to remove the luminescent substances. The fluorescein-free filter paper was soaked overnight in the CQD aqueous solution. The solvent was evaporated thoroughly to obtain the CQD-based test papers. Fe^3+^ solution and L-Cys were dripped onto the filter papers and a digital photo of the test papers was acquired in a UV light box (365 nm).

## 4. Conclusions

Food waste biomass was utilized to prepare CQDs as a simple, low-toxicity dual probe for Fe^3+^ and L-Cys detection. The CQDs were prepared from green pomelo peel via a one-step hydrothermal method using green pomelo peels as the precursor. After co-doped with N and S atoms, the resultant CQD solutions showed excellent optical properties, good fluorescence stabilities, and high QYs, up to 17.31%. The blue-emission CQD-based “turn-off–on” fluorescent probes with high selectivity and sensitivity were developed and employed for detection of Fe^3+^ and L-Cys. The detection limit for Fe^3+^ was 0.086 µM with a linear detection range of 0.1–160 µM, and the LOD for L-Cys was 0.34 µM with a linear detection range of 0.4–85 µM. Notably, common metal ions and amino acids did not interfere with the Fe^3+^ and L-Cys detection. These traits were attributed to the sorption of Fe^3+^ on the CQD surface and desorption of Fe^3+^ from the CQDs via the stronger bonding of L-Cys. Therefore, low-toxicity CQDs can be considered as promising candidates for visual optical sensing in practical sample analysis. Furthermore, cellulose/CQDs composite hydrogels and CQD-based test papers also showed good responsiveness to Fe^3+^ and L-Cys. The proposed CQD probes utilize a simple “turn-off–on” fluorescent mode, providing a high-efficiency platform for ideal analytical applications in food and environmental science.

## Data Availability

Not applicable.

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
