# Peer review of "Carbon Quantum Dots from Pomelo Peel as Fluorescence Probes for “Turn-Off–On” High-Sensitivity Detection of Fe3+ and L-Cysteine"

_molecules, 2022, doi:10.3390/molecules27134099_

Round 1

Reviewer 1 Report

The manuscript by Zhang et al. described using carbon quantum dots for fluorescent probe design. The overall design rationale is unclear in terms of analytical application. Details below:

  1. While the CQD and its manufacturing could be interesting to some readers, their “applications in sensing and analysis” is totally not justified. It is hard to imagine a real-world example where there is a dire need to first detect Fe3+ at ~50 uM, and then subsequently L-Cys at ~100 uM. The authors have mentioned that iron and cysteine are important species in biologically relevant systems, but there is no application of such at all. In fact, iron and cysteine almost always co-exist in the majority of biological systems such as cells or biofluids. In such a case, how will the designed probe respond to the mixture? Unless the probe could clearly report either species separately and quantitatively, the rationale for designing such a sequential probe is questionable. The ‘turn-off-on’ design is a strange design with very limited application.
  2. Considering the application scenario is unclear, the selectivity test could not be properly assessed. For example, if this probe is intended to be used inside a cell, then its selectivity against both Fe3+ and L-cys should be tested with a background of ~5 mM protein and ~10 mM glutathione that are present regardless of other conditions; in contrast, if the probe is intended to be used for wastewater measurement, then other heavy metal ions at high mM should be tested as an interfering species. The authors need to clarify the application scenario and test the selectivity for their designated analytes with normal background concentrations of other prevalent species.
  3. The fluorescent and absorbance signal as shown in Figures 3 and 4 both exhibit very limited changes upon specific analytes were added. What are the sensitivity requirements for the detection instrument? How about quantitation performance? The recovery showed quite poor results, indicating it is mostly a semi-quantitative method even in relatively clean background samples. If the only application is to detect the existence of a known additive to certain background, it is very hard to convince the readers the significance and usefulness of such research.
  4. The presentation of signal changes using F/F0 only in Figure 2 is not sufficient. The authors should present the raw traces of absorbance and fluorescence with proper control samples. Even with F/F0, the signal changes are still quite small as shown in Figure 3. This is one other caveat of using the ‘turn-off’ strategy as the signal change will be very subtle and prone to high background noise.
  5. Did the authors ever test other known binders of Fe3+ besides cysteine? It is very likely that pyridine, porphyrins, hemoglobin, other biothiols such as glutathione, H2S, etc. could all react similarly if not more reactive than cysteine for their probe design. In such a case, how could anyone attribute the signal change only to the cysteine but not other Fe3+ binders that are essentially everywhere?
  6. The authors have stated on page 7 line 209 using cysteine at 0.4-1000 mM while the other places such as figure legend suggest cysteine at 0.4-85 uM. Which one is true? There are also other inconsistent concentration reports in the manuscript, especially on cysteine concentration. If the probe design requires a high mM of cysteine, it further weakens the significance.

Reviewer 2 Report

The author presented a “turn-off-on” fluorescence sensor for detection of Fe3+ and L-cysteine based on CQDs from pomelo peel. I have some concerns that should be addressed.

  1. Line 100 – 101: the number of 100 nanoparticles is not small. However, to determine the average size more exactly and have a size distribution, the authors should carry out the DLS measurement.
  2. Figure 1E: Besides peaks of O1s, N1s, C1s and S2p, the XPS spectrum also shows the peaks located at around 500, 400, 350, and 300 eV. The authors must determine the peaks at these positions.
  3. The authors should smooth the raw line (black line) of the XPS spectra in the Figure 1G and Figure 1H.
  4. Line 141-145: The authors determine the emission wavelength of 410 nm. With this emission wavelength the authors should measure excitation scan to confirm the excitation peak (maximum excitation). The spectrum of excitation wavelength can show the maximum excitation more exactly.
  5. Figure 4B only shows the fluorescence spectrum of CQDs+Fe3++L-Cys. There are no spectra of CQDs and CQDs + Fe3+.
  6. Figure 5 (First image): The authors must put the test papers on a clear background and then take a picture.
  7. What is the method (formular) the authors used to determine the LOD of Fe3+, and the LOD of L-Cys?
  8. The authors must compare the detection results of Fe3+ and L-Cys with those of the different methods.
  9. L-Cysteine, Homocysteine, and Glutathione belong to biological thiol family, which have same functional groups. Did the authors check with homocysteine and glutathione? I wonder they can show the same phenomenon to that of L-Cysteine.

Round 2

Reviewer 1 Report

The authors did not address most of the questions raised and thus the revised manuscript should not be published. Detailed responses are below:

1.       Regarding sensor design using an ‘off-on’ mechanism, citing previous literature using a similar strategy does not justify the probe design issue here. The authors claimed that the intended application was to use in tap water and then amino acid drinks, does that mean that the probe could only be used in such sequences in which the users first need to put the probe in tap water then in the drinks? If there is not a sufficient amount of Fe3+ in the tap water, then the users could not use the subsequent step? What is the difference between using two separated sensors in two separate media using this proposed probe? How about if Fe3+ and L-Cys co-exist in the tap water? What will be the results?

2.       Regarding selectivity, it is clear that some ions such as Cu2+, Mg2+ and Al3+ could clearly change the fluorescent response, albeit not as strong. In the response letter, the authors showed that 2 folds of Cu2+/Mg2+/Al3+ could decrease the blank signal from 0.25 to 0.15, a ~40% change. This change is not small/neglectable. If Fe3+ and these ions co-exist in the sample, the signal could be the mixed response of all of them. How does the user know if their signal is truly from Fe3+ but not other ions?

3.       Similarly, regarding selectivity against L-cys, the authors did not test the suggested Fe3+ binders such as pyridine, porphyrins, hemoglobin. It is not acceptable just claiming that they ‘think’ these compounds will not affect the sensing without showing any data. These compounds are commonly present in many samples ranging from natural water to biofluids. The authors also did not specify the concentrations that they use to test biothiols selectivity and did not check H2S. The current data does not support their claim of specificity.

4.       The authors did not update the manuscript with the revision experiment results. 

Reviewer 2 Report

As we suggested the authors revised the manuscript properly.

Author Response

Thank you very much.